# Learning Object-Centric Representation via Reverse Hierarchy Guidance

## Abstract

Object-Centric Learning (OCL) seeks to enable Neural Networks to identify individual objects in a visual scene in an unsupervised manner, which is a meaningful task because the ability to recognize objects and understand their relationships is the foundation of interpretable visual comprehension and reasoning. Due to humans' strong ability to split visual scenes into object sets, incorporating the mechanism of human visual perception into model architecture is a potential way to enhance object representation. According to Reverse Hierarchy Theory (RHT), the human visual system comprises two reverse processes: a bottom-up process rapidly extracting the gist of scenes and a top-down process integrating detailed information into consciousness. Inspired by RHT, We propose Reverse Hierarchy Guided Network (RHGNet) that enhances the models' object-centric representations through an extra top-down pathway as described in RHT. This pathway allows for more decisive semantic information to be included in extracted low-level features, as well as helps search for optimal solutions to distinguish objects from low-level features. We demonstrate with experiments that the model benefits from our method and achieves a stronger ability to differentiate objects, especially the easily ignored small and occluded ones, than current models following a pure bottom-up fashion.

## 1 Introduction

The human visual system is skilled at parsing visual scenes into object compositions (Kahneman et al., 1992). This property provides an efficient and interpretable representation of visual scenes. As a result, Object-Centric Learning (OCL) may be a promising research direction for improving the interpretability and human-like properties of Neural Networks. In OCL, networks decompose an image into a set of objects (often represented by latent vectors referred to as 'slots') (Greff et al., 2019; Locatello et al., 2020; Burgess et al., 2019). Each slot reconstructs its corresponding object through a decoder network.

Previous methods have already achieved considerable object-centric representations in their models. However, object-level mistakes are still repetitively witnessed, such as an object missing in the reconstructed picture or two nearby objects reconstructed as a single object. Such mistakes are more common in small objects, particularly those partially occluded by surrounding things. Fig.1(b) shows examples of missing objects. This problem is partly caused by the loss of information with low saliency when models encode low-level features into slots. Considering that OCL models compress information from numerous pixels to several slots, small and occluded objects occupying fewer pixels are more likely to be viewed as redundant information and discarded.

Reverse Hierarchy Theory(RHT)(Hochstein & Ahissar, 2002), a visual theory used to explain human visual phenomena, provides a feasible way to address such problems. RHT argues that the human visual system consists of two pathways. The bottom-up pathway performs a fast vision for analyzing the gist of the visual scene. During the bottom-up process, only salient information is transmitted to higher-level neurons, and detailed information is discarded. This mechanism makes the bottom-up process very rapid. However, it also leads to vision errors because visual attention is guided by saliency, while secondary objects, such as small and occluded ones, may be ignored. Suppose it is necessary to correct these errors. In that case, the top-down pathway will be started by returning to a

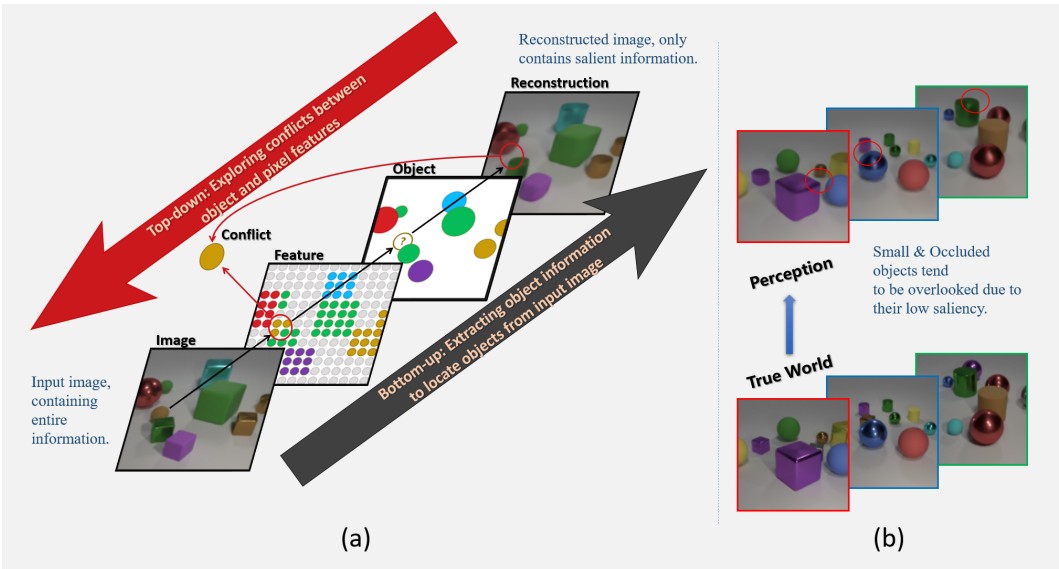

Figure 1: An abstract of the Reverse Hierarchy Guidance. **(a)** Transferring RHT structure into the neural network design. Natural visual perception follows a bottom-up manner, during which the model successively extracts low-level features and clusters them into objects. In this process, small and occluded objects are often ignored because they are of low saliency. The top-down pathway is designed to retrieve these missing objects by detecting conflicts between objects and low-level features. The conflicts help optimize the model in two ways: either work on improving low-level vision by enhancing the feature of missing objects, or work on high-level features, improving the quality of the clustering process. **(b)** Examples of missing low-saliency objects in bottom-up models.

lower level of awareness and bringing detailed information into consciousness, which validates the bottom-up perception.

Inspired by RHT, we suggest Reverse Hierarchy Guided Network (RHGNet), which employs an additional top-down pathway to improve object-centric representations of visual scenes. This pathway detects conflicts where an object exists in low-level features but is ignored in high-level object perception. Specifically, a bottom-up model similar to previous OCL models (Locatello et al., 2020; Jia et al., 2023; Chang et al., 2022) is first utilized to extract low-level features and encode them into slots. Then the model produces object segmentations in two ways: a high-level segmentation by decoding slots with a Spatial Broadcast Decoder (Watters et al., 2019) and a low-level segmentation by assigning low-level features to slots with a top-down attention module. High- and low-level segmentations should be highly consistent if the result is correct. If an object is missed, a conflict emerges between the two segmentations. In our approach, the role of the top-down pathway is to solve such conflicts. This process is presented in different ways during training and inference, both of which help improve the ability of the network to distinguish objects.

We evaluate our proposed RHGNet and compare it with current SOTA models on several datasets, including CLEVRTex (Karazija et al., 2021), CLEVR (Johnson et al., 2017), and ObjectsRoom (Kabra et al., 2019). The experiment results indicate that the more challenging the dataset, the more RHGNet outperforms other models. Using the common ARI-FG (Rand, 1971) as evaluation metrics, RHGNet performs better than other SOTA models on all the datasets, significantly outperforming other models by a large margin on CLEVRTex, the most challenging dataset we use. In addition, we evaluate models' performance on objects of different sizes with an IoU-based metric to demonstrate that the most significant boost occurs on small objects. Our method allows the model to scrutinize the scene, thus finding objects missing in the bottom-up process and reinforcing the object-centric representations. In addition, we show the performance of our model on two real-world datasets, i.e., MOVi-C and COCO, demonstrating that our method can be combined with existing methods to generalize to real-world scenarios and achieve better performance.

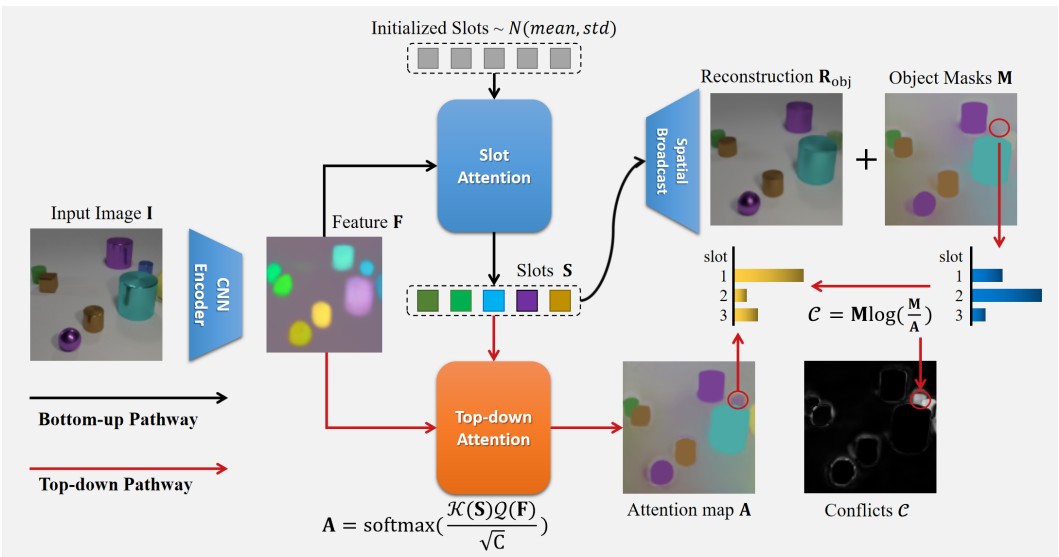

Figure 2: **Overview**. Current models follow a pure bottom-up fashion (Locatello et al., 2020; Jia et al., 2023; Chang et al., 2022), producing a reconstruction result and object masks. An extra top-down pathway is applied in our model, which introduces a Top-down Attention module to compute an attention map between low-level features and slots. A conflict is computed between the object masks and the top-down attention map with KL Divergence(Kullback, 1997). The conflict acts as a guidance signal to help optimize the low-level feature during training and search for appropriate initial slots that perform reliable clustering during inference.

## 2   RELATED WORK

**Object-Centric Learning.** OCL aims at enabling Neural Networks to perceive environments in a similar way to human vision. Mainstream OCL models often focus on a basic property, i.e., discovering objects in visual scenes. Earlier works, including IODINE (Greff et al., 2019), MONet (Burgess et al., 2019) and GENESIS (Engelcke et al., 2019), accomplish this task by using multiple encoder-decoder structures. Slot-Attention (Locatello et al., 2020), instead, proposed an iterative attention method that allows slots to compete for input image segments and conduct segmentation parallelly, requiring one time of encoding computation. A critical issue faced by OCL research is about how to generalize to complex, real-world scenes. SLATE (Singh et al., 2021) introduces a transformer-based decoder to enhance the model's reconstruction ability. DINOSAUR (Seitzer et al., 2022) argues that the simple reconstruction task is insufficient to distinguish objects and achieve better performance by changing the reconstruction objective to the output feature of DINO (Caron et al., 2021). BO-QSA (Jia et al., 2023) and I-SA (Chang et al., 2022) focus on Query Optimization, which uses learnable parameters to initialize slots instead of random sampling.

**Reverse Hierarchy Theory.** RHT (Hochstein & Ahissar, 2002; Ahissar & Hochstein, 2004) is introduced to explain the visual phenomena of humans. It claimed that there are two opposite pathways in the human visual system. The bottom-up pathway works implicitly and acquires the gist of the scene rapidly. However, most detailed information is left out in this process, which may cause errors or blindness in perception. For more precise perceptions, the top-down pathway is initiated consciously by returning to lower-level neurons through feedback connection, bringing detailed information into consciousness and correcting possible errors. RHT offers a valid explanation of the human vision mechanism, which is consistent with a series of visual or biological experiments and theories (Potter, 1976; Rensink et al., 1997; Kanwisher, 1987; Juan & Walsh, 2003; Wolfe, 2021). The thought of adding a top-down pathway in the network has been widely adopted in other fields of computer vision, e.g., semantic segmentation (Yin et al., 2022), visual saliency (Ramanishka et al., 2017) and vision question & answering (Anderson et al., 2018). It remains under research how much improvement RHT may bring to the field of OCL, and more importantly, how RHT can benefit the interpretability of models.

## 3 METHOD

### 3.1 BOTTOM-UP PATHWAY

The overall architecture of RHGNet is shown in Fig.2. A model with similar architecture to previous OCL models (Locatello et al., 2020; Jia et al., 2023; Chang et al., 2022) is adopted as our bottom-up model. The input image $\mathbf{I} \in \mathbb{R}^{B \times 3 \times H \times W}$ first passes through a CNN network to get low-level features $\mathbf{F} \in \mathbb{R}^{B \times C \times H/s \times W/s}$. Then a Slot-Attention module (Locatello et al., 2020) is employed to encode $\mathbf{F}$ into $K$ slots $\mathbf{S} \in \mathbb{R}^{B \times K \times C}$. Finally, with a Spatial Broadcast Decoder $\mathcal{SBD}$ (Watters et al., 2019), each slot is reconstructed into its corresponding object in the image in the form of an RGB reconstruction $\mathbf{R} \in \mathbb{R}^{B \times K \times 3 \times H \times W}$ and an object mask $\mathbf{M} \in \mathbb{R}^{B \times K \times 1 \times H \times W}$. The final reconstruction result $\mathbf{R}_{\text{obj}}$ is the weighted sum of $\mathbf{R}$ and $\mathbf{M}$:

$$\begin{cases} \mathbf{R}, \mathbf{M} = \mathcal{SBD}(\mathbf{S}), \\ \mathbf{R}_{\text{obj}} = \text{sum}(\mathbf{R} * \mathbf{M}, \text{axis} = \text{slots}). \end{cases} \tag{1}$$

### 3.2 REVERSE HIERARCHY GUIDANCE AS TOP-DOWN PATHWAY

The bottom-up pathway conducts a strong information compression to express numerous pixels with several slots, which may cause blindness due to the discarded detailed information. The top-down pathway is designed to overcome these flaws through re-entry to low-level features where the discarded information is available. Formally, given low-level features $\mathbf{F}$ and object slots $\mathbf{S}$, we define a conflict function $\mathcal{C}(\cdot, \cdot)$ to quantize how $\mathbf{S}$ is inconsistent with $\mathbf{F}$. For the areas successfully reconstructed, $\mathbf{F}$ and $\mathbf{S}$ have high consistency and $\mathcal{C}(\cdot, \cdot)$ should return a low value. On the contrary, if a certain object is ignored, $\mathcal{C}(\cdot, \cdot)$ becomes large in the corresponding area. The top-down pathway is modeled as the process of minimizing the quantized conflicts:

$$\min \mathcal{C}(\mathbf{F}, \mathbf{S}). \tag{2}$$

Because $\mathbf{F}$ is expressed as a matrix, while $\mathbf{S}$ is a set, they cannot be compared directly. Instead, we incorporate an intuitive idea that "a pixel should be reconstructed by the slot that is most similar to it". Specifically, we use a top-down attention module to compute the similarity between low-level features and slots, and compare it with the object masks $\mathbf{M}$. Formally, we define trainable linear projections $\mathcal{Q}(\cdot), \mathcal{K}(\cdot)$. The target dimension of the projections remains $C$. The top-down attention $\mathbf{A}$ is computed by:

$$\mathbf{A} = \text{softmax}(\frac{\mathcal{K}(\mathbf{S})\mathcal{Q}(\mathbf{F})^{\text{T}}}{\sqrt{\text{C}}}, \text{axis} = \text{slots}). \tag{3}$$

Considering both top-down attention map $\mathbf{A}$ and object masks $\mathbf{M}$ are in the form of a normalized probability distribution, KL divergence $\mathcal{KL}(\cdot||\cdot)$ (Kullback, 1997) is utilized to measure the difference between them. The conflict between low-level features and slots is defined as

$$\mathcal{C}(\mathbf{F}, \mathbf{S}) := \mathcal{KL}(\mathbf{M}||\mathbf{A}) = \sum \mathbf{M}\log(\frac{\mathbf{M}}{\mathbf{A}}), \tag{4}$$

where $\mathbf{A}$ is upsampled to the same resolution as $\mathbf{M}$ before computing $\mathcal{C}$. In our method, the quantified conflicts are optimized differently during training and inference.

**Training-Time Reverse Hierarchy Guidance (Train-RHG).** During training, the conflicts will be expressed as a loss function termed Top-down Conflict Loss ($\mathbf{L}_{\text{TDC}} := \frac{1}{HW} \sum_{H,W} \mathcal{KL}(\mathbf{M}||\mathbf{A})$). $\mathbf{L}_{\text{TDC}}$ guides the model to learn more semantic features by clustering the features belonging to the same object while separating those belonging to different ones. We introduce Object Reconstruction Loss $\mathbf{L}_{\text{obj}} := \|\mathbf{R}_{\text{obj}} - \mathbf{I}\|_1$ to help generate initial object-centric representation. In addition, an extra shallow CNN decoder $\mathcal{D}$ is introduced to reconstruct the input image from $\mathbf{F}$. The Information Integrity Loss $\mathbf{L}_{\text{info}} := \|\mathcal{D}(\mathbf{F}) - \mathbf{I}\|_1$ ensures that all the necessary detailed information is available in low-level features.

The final loss $\mathbf{L}$ is the weighted sum of the three losses mentioned above:

$$\mathbf{L} = \mathbf{L}_{\text{obj}} + \lambda_{\text{info}}\mathbf{L}_{\text{info}} + \lambda_{\text{TDC}}\mathbf{L}_{\text{TDC}}. \tag{5}$$

**Inference-Time Reverse Hierarchy Guidance (Infer-RHG).** The Slot-Attention module generates initial slots through random sampling. Although this may bring some remarkable properties, e.g., permutation invariance, extreme initial values of slots deteriorate the segmentation results significantly. As shown in Fig.3(a), we repeat running RHGNet 100 times, recording the model performance (using ARI-FG (Rand, 1971) as metrics) and the top-down conflict. Model performance varies over different runnings, with inferior results rating approximately 20% lower than superior ones.

A clear negative correlation exists between model performance and top-down conflicts, where results with lower conflicts are more likely to achieve high model performance. Based on this observation, we propose that the top-down pathway generates a signal to guide the initialization of slots, leading to more consistent segmentation results. After the bottom-up model produces object masks, the top-down pathway is initiated to compute conflict $\mathcal{C}$. We repeat running the bottom-up and top-down processes for $N$ times and choose a result with the smallest conflict. $N$ is set to 10 in our experiment, which is enough to provide considerable performance gains.

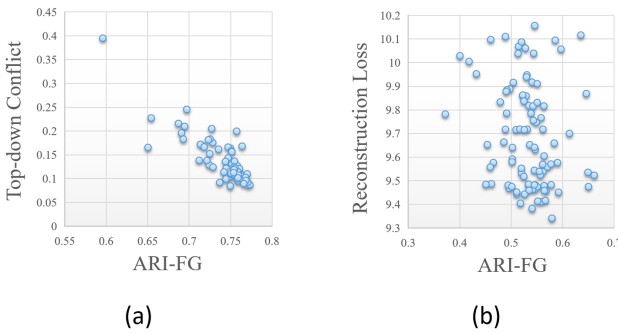

Figure 3: Relationship between guidance signals and model performance during inference. **(a)** RHGNet, using conflicts as guidance. **(b)** bottom-up model, using reconstruction loss as guidance. The conflict's relevance to ARI-FG in RHGNet is much stronger than that of reconstruction loss in a bottom-up model.

We also tried other possible schemes to achieve Infer-RHG. First, we verified whether the reconstruction loss could be used as the guide signal instead of the top-down conflict. As is shown in Fig.3(b), there is no apparent negative correlation between the reconstruction loss and the model performance, indicating that Infer-RHG should work based on our Reverse Hierarchy network design. In addition, we tried to use back-propagation and simulated annealing algorithms (Kirkpatrick et al., 1983) to optimize the initial slot value, finding that it requires too many iterations to search for a proper initial value, significantly increasing the computation cost, which is not suitable for the inference process. Related experiment results are available in the supplementary material.

# 4 EXPERIMENTS

## 4.1 IMPLEMETATION DETAILS

**Datasets.** We use CLEVR(Johnson et al., 2017) and ObjectsRoom from multi_object_datasets(Kabra et al., 2019), and CLEVRTex (Karazija et al., 2021) for training and evaluation of our model. Dataset split follows (Jia et al., 2023). CLEVRTex-CAMO and CLEVRTex-OOD, two test sets for CLEVRTex, are also used to evaluate our model to test the model's generalization ability. For CLEVR and CLEVRTex, we crop the center 192*192 pixels and resize the image to 128*128; for ObjectsRoom, the 64*64 images are directly used as input. The RGB values are normalized to [-1,1]. We also evaluate our method on two real-world datasets, i.e., MOVi-C (Greff et al., 2022) and COCO (Caesar et al., 2018) to show that our methods can generalize well on complex, real-world scenes. Real-world images are resized to 224*224 before input into the network.

**Network Architecture.** A ResNet18 encoder is used to acquire low-level features $\mathbf{F} \in \mathbb{R}^{B \times C \times H/s \times W/s}$. Here $C = 64$. $s$ is set to 1 for ObjectsRoom and 2 for CLEVR and CLEVRTex. The Slot-Attention module encodes low-level features into slots $\mathbf{S} \in \mathbb{R}^{B \times K \times C}$. $K$ is set to 7 for ObjectsRoom and 11 for CLEVR and CLEVRTex. A Spatial Broadcast Decoder (Watters et al., 2019) generates the reconstruction results and object masks from slots. In the top-down pathway, the top-down attention is computed between low-level features and slots, acquiring a normalized attention map, which is then upsampled to a resolution the same as object masks through bilinear interpolation to compute the conflict. To transfer our method to real-world datasets, we follow the

thought of DINOSAUR (Seitzer et al., 2022) by replacing the input and the reconstruction objective of the model with pre-trained DINO features (**?**). More detailed settings on real-world datasets are available in supplementary material.

**Loss Weight.** $\lambda_{\mathrm{info}}$ is set to 0.1. $\lambda_{\mathrm{TDC}}$ is set to 0 in the first $\frac{1}{4}$ steps and gradually increases to $\lambda_{\mathrm{TDC_{max}}}$ from $\frac{1}{4}$ to $\frac{3}{4}$ steps in a cosine manner, keeping its value in the rest steps. $\lambda_{\mathrm{TDC_{max}}}$ is set to 0.1 on CLEVR and ObjectsRoom, while 1.0 on CLEVRTex and real-world datasets.

**Metrics.** We use foreground adjusted rand index (ARI-FG) (Rand, 1971) to evaluate the models' ability to discover objects. Following previous works (Karazija et al., 2021; Seitzer et al., 2022), we also introduce mean best overlapping (mBO) on real-world datasets and mean square error (MSE) on synthetic datasets for evaluation

Although ARI-FG is commonly used for evaluating OCL models, it is not a balanced metric because bigger objects have larger weights than smaller ones. To address this problem, we employ an IoU-based metric called Object IoU(OIoU). OIoU is similar to mIoU used in supervised segmentation tasks. The difference is that an extra KM algorithm (Kuhn, 1955; Munkres, 1957) matches between model predictions and ground truth. In OIoU, each item is assigned a predicted mask to

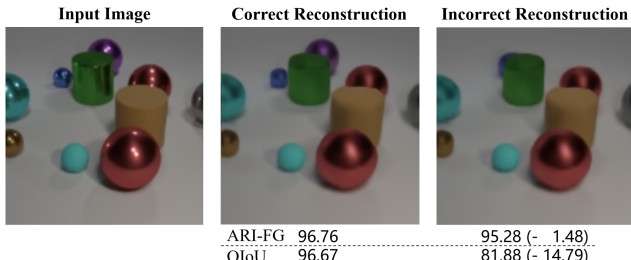

Figure 4: An example showing OIoU reflects missing small objects better than ARI-FG.

compute an IoU score ranging from 0% to 100%, allowing us to study the relationship between models' object discovery ability and object size. Fig.4 illustrates the comparison between the two metrics. In the incorrect reconstruction result, the small blue ball and large purple sphere are assigned to the same slot, generating a blue sphere in the middle. This error only leads to a 1.48% ARI-FG decrease. In contrast, OIoU decreases by 14.79%, correctly reflecting object-missing situations.

**Hyperparameter.** We train all our models with an AdamW (Loshchilov & Hutter, 2017) optimizer and a batch size of 64 for 400k steps. The learning rate begins with $4 \times 10^{-4}$ and decreases to $1 \times 10^{-5}$ at the end of the training through a CosineAnnealing strategy. Warm-up is not used. The training of our model takes about one day on 4 GeForce GTX TITAN XP graphic cards.

Table 1: Model performance on CLEVR and CLEVRTex, comparing with current SOTA models. CAMO and OOD represent CLEVRTex-CAMO and CLEVRTex-OOD. More detailed comparisons with the standard deviation of model performance are available in supplementary material. Models marked with red font are newly added during discussion.

| model | CLEVRTex | | CAMO | | OOD | | CLEVR | |
|---|---|---|---|---|---|---|---|---|
| | ↑ARI-FG | ↓MSE | ↑ARI-FG | ↓MSE | ↑ARI-FG | ↓MSE | ↑ARI-FG | ↓MSE |
| IODINE (Greff et al., 2019) | 59.52 | 340 | 36.31 | 315 | 53.20 | 504 | 93.81 | 44 |
| DTI (Monnier et al., 2021) | 79.90 | 438 | 72.90 | 377 | 73.67 | 590 | 89.54 | 77 |
| eMORL (Monnier et al., 2021) | 45.00 | 318 | 42.34 | 269 | 43.13 | 471 | 93.25 | 26 |
| MONet (Burgess et al., 2019) | 36.66 | 146 | 31.52 | **112** | 37.29 | 231 | 54.47 | 58 |
| GEN-v2 (Engelcke et al., 2021) | 31.19 | 315 | 29.60 | 278 | 29.04 | 539 | 57.90 | 158 |
| SLATE (Singh et al., 2021) | 45.44 | 498 | 43.52 | 349 | 46.49 | 550 | - | 50 |
| SA (Locatello et al., 2020) | 62.40 | 254 | 57.54 | 215 | 58.45 | 487 | 95.89 | 23 |
| I-SA (Chang et al., 2022) | 78.96 | 280 | 72.25 | 271 | 73.78 | 515 | - | 11 |
| BO-QSA (Jia et al., 2023) | 80.47 | 268 | 72.59 | 246 | 72.45 | 805 | 96.90 | 17 |
| AST-Seg (Sauvalle & de La Fortelle, 2022) | 71.79 | 152 | - | - | - | - | 76.05 | 51 |
| SA+SLP (Chakravarthy et al., 2023) | 71.00 | - | - | - | - | - | - | - |
| BO-QSA+SLP (Chakravarthy et al., 2023) | 87.00 | - | - | - | - | - | - | - |
| RHGNet (ours) | 89.53 | 120 | 81.78 | 116 | 79.58 | 226 | 98.31 | 9 |
| +Infer-RHG | **89.90** | **118** | **82.85** | 121 | **80.25** | **222** | **98.55** | **8** |

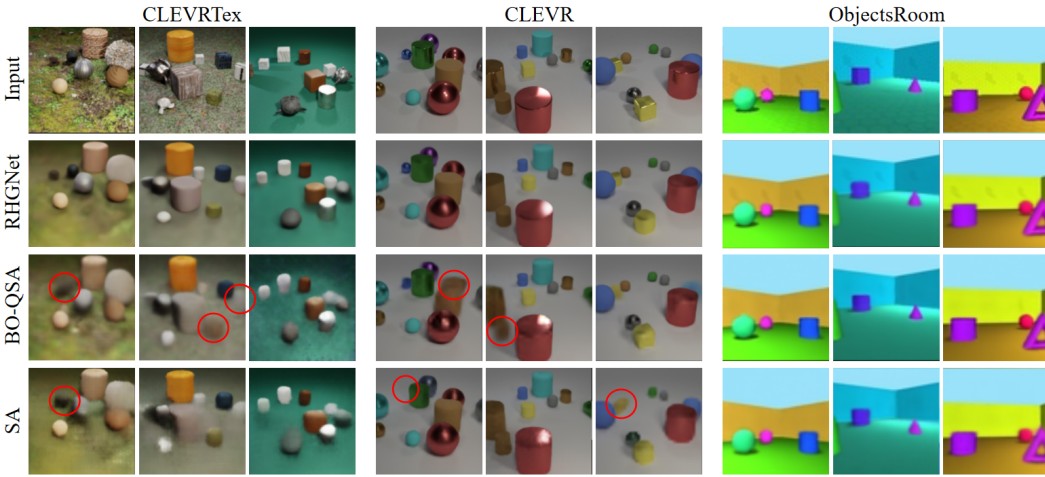

Figure 5: A comparison between our proposed RHGNet, SA (Locatello et al., 2020) and BO-QSA (Jia et al., 2023) on CLEVRTex, CLEVR and ObjectsRoom.

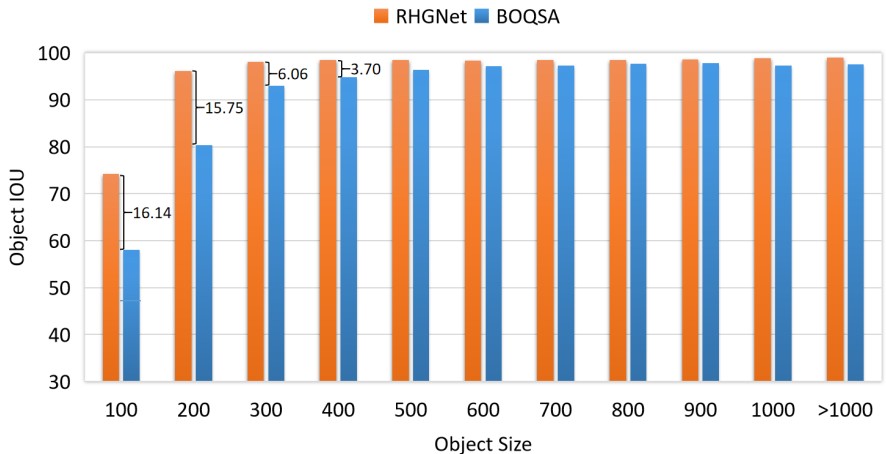

Figure 6: Models' object discovery ability on CLEVR. We compare RHGNet and BO-QSA on various sizes of objects in terms of OIoU, confirming that RHGNet performs well on small objects.

## 4.2 QUANTITATIVE RESULTS

We compare the performance of RHGNet and other SOTA models on CLEVRTex, CLEVR, and ObjectsRoom. To measure the effects of Train-RHG and Infer-RHG, we adopt two inference ways: one that only initiates the bottom-up model and another that applies the Infer-RHG to enhance the performance further. Experiment results are shown in Tab.1. RHGNet, with only bottom-up inference, already outperforms current models on most metrics. Infer-RHG further provides a stable performance boost. As the difficulty of the dataset increases, RHGNet shows more obvious advantages, with **1.65%** and **2.90%** ARI-FG higher than other best models on CLEVR and CLEVRTex. Our model also generalizes well on CLEVRTex-CAMO and CLEVRTex-OOD, outperforming other models by a large margin. The MSE of reconstructed images is also lower than other models.

The visualization results are shown in Fig.5, which compares the performance of RHGNet, SA, and BO-QSA. ObjectsRoom is the simplest of the three datasets. All of these models are capable of producing precise reconstruction results. On CLEVR, RHGNet outperforms SA and BO-QSA, which make mistakes when generating small and occluded objects. The areas of these errors are circled in the figure. CLEVRTex is the most challenging dataset because the texture information is hard

Table 2: Model performance comparison on real-world datasets MOVi-C and COCO.

| model | MOVi-C | | COCO | |
|---|---|---|---|---|
| | ↑ARI-FG | ↑mBO | ↑ARI-FG | ↑mBO |
| Slot-Attention | 43.8±0.3 | 26.2±1.0 | 16.4±3.6 | 14.7±1.0 |
| SLATE | 43.6±1.3 | 26.5±1.1 | 24.1±0.2 | 19.9±0.1 |
| DINOSAUR | 67.2±0.3 | 38.6±0.1 | 40.5±0.0 | 27.7±0.2 |
| RHGNet | 70.50±0.3 | 39.64±0.2 | 41.02±0.5 | 27.90±0.4 |
| +Infer-RHG | **73.00±0.3** | **40.68±0.2** | **41.14±0.4** | **28.26±0.4** |

to rebuild and may interfere with the model's judgment when segmenting objects. Other models show failure cases on objects of various sizes and generate invalid reconstructions. RHGNet, instead, shows its capacity to retain crucial information, succeeding in distinguishing individual objects from their surroundings and avoiding errors on small objects.

Using OIoU, we compared the ability of our model to discover objects of different sizes with BO-QSA (Jia et al., 2023). As is shown in Fig.6, compared to BO-QSA, RHGNet's ability to discover small objects is greatly improved on CLEVR.

In Tab.2, we compare the performance of our model with the existing model in a real-world scenario. On both MOVi-C and COCO datasets, we achieved higher performance with smaller or comparable models. Compared with DINOSAUR with the same pre-trained backbone, Train-RHG and Infer-RHG increase ARI-FG by 3.30% and 2.50% on MOVi-C, as well as 0.52% and 0.12% on COCO.

### 4.3 ABLATION STUDIES: SMALL OBJECT DISCOVERY THROUGH REVERSE HIERARCHY GUIDANCE

Ablative experiments are conducted on CLEVR and CLEVRTex. Experiment results are listed in Tab.3. In the experiments, the objects are divided into three sizes according to the number of pixels they occupy: small objects occupy less than 150 pixels, large objects occupy more than 1000 pixels, and the rest are middle size. We analyze three factors applied in our network design through ablative experiments: the Information Integrity Loss $\mathbf{L}_{\text{info}}$ (**Info** in Tab.3), the Top-Down Conflict Loss $\mathbf{L}_{\text{TDC}}$ (**TDC** in Tab.3), and the Inference-time Reverse Hierarchy Guidance (**Infer-RHG** in Tab.3).

We compute models' ARI-FG and OIoU scores on objects of various sizes (OIoU-S, -M, and -L represent OIoU on small, middle, and large objects). On both datasets, RHGNet outperforms BUNet vastly in the segmentation of small objects, with **11.70%** and **9.21%** OIoU-S improvement respectively on CLEVRTex and CLEVR. The segmentation of larger objects also benefits from our method on CLEVETex, receiving an improvement of approximately **10%** OIoU. We conclude that RHGNet overcomes more challenging samples that a bottom-up model can not solve. In simpler datasets, these samples mainly focus on small or occluded objects, while in more difficult datasets, the scope is extended to objects of all sizes.

Table 3: Quantitative result of ablation studies.

| Ablated factors | | | CLEVRTex | | | | CLEVR | | | |
|---|---|---|---|---|---|---|---|---|---|---|
| Info | TDC | Infer-RHG | ↑ARI-FG | ↑OIoU-S | ↑OIoU-M | ↑OIoU-L | ↑ARI-FG | ↑OIoU-S | ↑OIoU-M | ↑OIoU-L |
| - | - | - | 78.15 | 47.21 | 78.77 | 83.31 | 97.17 | 78.26 | 94.72 | 97.77 |
| ✓ | - | - | 86.92 | 49.99 | 84.43 | 91.25 | 97.62 | 78.49 | 95.79 | 98.26 |
| - | ✓ | - | 87.18 | 50.82 | 85.22 | 91.68 | 97.97 | 83.73 | 96.71 | 98.52 |
| ✓ | ✓ | - | 89.53 | 56.24 | 87.47 | 93.82 | 98.31 | 85.00 | 97.17 | 98.73 |
| ✓ | ✓ | ✓ | **89.90** | **58.91** | **88.76** | **93.88** | **98.55** | **87.47** | **98.02** | **98.89** |

To analyze how Reverse Hierarchy Guidance benefits the model, we visualize the internal features and attentions of different models in Fig.7, including reconstructions, object masks, attention maps in the Slot-Attention module, top-down attention maps, and low-level features (acquired by reducing

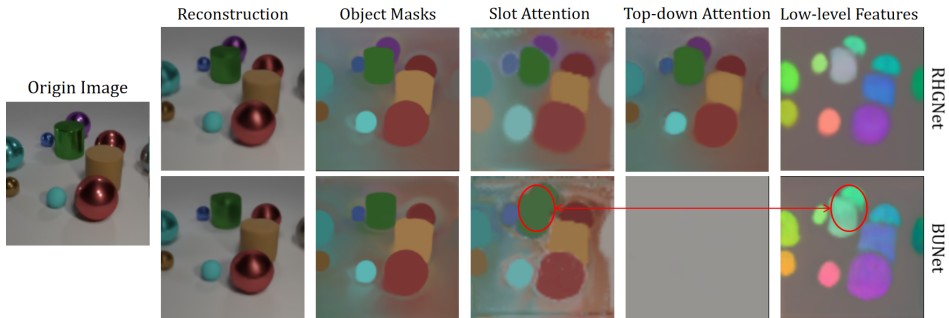

Figure 7: The internal visualized results of RHGNet compared with BUNet, showing bad cases of the bottom-up model on occluded objects that occupy few pixels. The confused low-level features of BUNet cause such errors, as is circled in the figure.

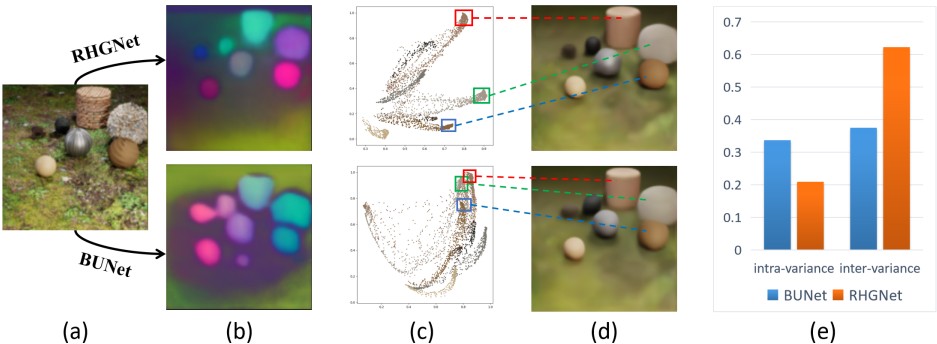

Figure 8: The visualized results on CLEVRTex. **(a)** the input image. **(b)** the low-level features with dimensions reduced to 3. **(c)** low-level features plotted on a 2D plane. PCA (Abdi & Williams, 2010) is adopted to reduce feature dimensions in (b) and (c). **(d)** the reconstruction results. Part of the objects and their corresponding features in (c) are marked with red, blue, and green lanes. **(e)** the statistics of intra- and inter-object feature variance of different models.

feature dimension to 3 with PCA (Abdi & Williams, 2010)). **BUNet** in the figure stands for the pure bottom-up model. In BUNet, the features of close objects are similar due to position embedding, making them more easily assigned to the same slot, reflected as missing objects. Reconstruction loss is insufficient for teaching the model enough semantic information. In RHGNet, definite boundaries separate nearby objects because low-level features receive guidance from high-level object masks, keeping them close to their corresponding slot and away from the others.

This property is also investigated on CLEVRTex. As illustrated in Fig.8, RHGNet extracts more identifiable features than BUNet, with three nearby objects owning separated features as marked in Fig.8 (c) and (d). The quantitative result from Fig.8(e) confirms that RHGNet achieves a higher inter-object feature variance and a lower intra-object feature variance.

## 5  CONCLUSION

Drawing lessons from RHT, we propose RHGNet, which constructs a connection between high-level object representations (i.e., slots) and low-level features through the conflict between them. We demonstrated that our model helps discover small or occluded objects, as well as achieves significant performance improvements on more complex scenarios, e.g., CLEVRTex or real-world datasets. Such improvement comes from two advantages brought about by Reverse Hierarchy Guidance. (i) It helps the model distinguish objects in low-level perception, represented by higher inter-object feature variance and lower intra-object feature variance. (ii) In the process of clustering low-level features into slots, it helps search for better initial slots, thus providing more stable clustering results.

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
