# Learning Object-Centric Representation via Reverse Hierarchy Guidance
# Supplementary Material

## A  Model Performance with Standard Deviation

Limited by space, we did not report the deviation of the models' performances. Here, we provide the full model performance table with the standard deviation of model performance calculated over 3 runs. The results are shown in the form of "$mean \pm std$". Tab.1 records the model performance on CLEVRTex, CLEVRTex-CAMO, and CLEVRTex-OOD, while the model performance on CLEVR and ObjectsRoom is available in Tab.2.

Table 1: Full model performance on CLEVRTex, CLEVRTex-CAMO, and CLEVRTex-OOD. Models with 0 performance standard deviation mean we use the model weight from their official repository. Models marked with red font are newly added during discussion.

| model | CLEVRTex | | CAMO | | OOD | |
|---|---|---|---|---|---|---|
| | ↑ARI-FG | ↓MSE | ↑ARI-FG | ↓MSE | ↑ARI-FG | ↓MSE |
| IODINE (Greff et al., 2019) | 59.52±2.20 | 340±3 | 36.31±2.57 | 315±3 | 53.20±2.55 | 504±3 |
| DTI (Monnier et al., 2021) | 79.90±1.37 | 438±22 | 72.90±1.89 | 377±17 | 73.67±0.98 | 590±4 |
| eMORL (Monnier et al., 2021) | 45.00±7.77 | 318±43 | 42.34±7.19 | 269±31 | 43.13±9.28 | 471±51 |
| MONet (Burgess et al., 2019) | 36.66±0.87 | 146±7 | 31.52±0.73 | **112±7** | 37.29±1.00 | 231±7 |
| GEN-v2 (Engelcke et al., 2021) | 31.19±12.41 | 315±106 | 29.60±12.84 | 278±75 | 29.04±11.23 | 539±147 |
| SLATE (Singh et al., 2021) | 45.44±5.16 | 498±12 | 43.52±4.32 | 349±9 | 46.49±5.44 | 550±14 |
| SA (Locatello et al., 2020) | 62.40±2.23 | 254±8 | 57.54±1.01 | 215±7 | 58.45±1.87 | 487±16 |
| I-SA (Chang et al., 2022) | 78.96±3.88 | 280±8 | 72.25±2.25 | 271±7 | 73.78±3.41 | 515±11 |
| BO-QSA (Jia et al., 2023) | 80.47±2.49 | 268±2 | 72.59±0 | 246±0 | 72.45±0 | 805±0 |
| AST-Seg-B3-BT (Jia et al., 2023) | 71.79±22.88 | 152±39 | - | - | - | - |
| SA+SLP (Jia et al., 2023) | 71.00±5 | - | - | - | - | - |
| BO-QSA+SLP (Jia et al., 2023) | 87.00±5 | - | - | - | - | - |
| RHGNet (ours) | 89.53±0.46 | 120±4 | 81.78±1.12 | 116±3 | 79.58±1.01 | 226±8 |
| +Infer-RHG | **89.90±0.42** | **118±4** | **82.85±1.06** | 121±4 | **80.25±0.93** | **222±8** |

## B  Searching Methods of Infer-RHG

Inference-time Reverse Hierarchy Guidance (Infer-RHG) is introduced to search for appropriate initial slots in the paper. To achieve the search, we choose a search method adapted from the simulated annealing algorithm, where the temperature starts from a small value ($10^{-8}$). We also studied other possible searching methods, including the original simulated annealing and the back-propagation algorithm. In the original simulated annealing algorithm, the temperature starts from $10^{-2}$ and decreases to $10^{-8}$ in a cosine manner. In the back-propagation algorithm, we use an SGD optimizer with a learning rate of 0.1 to optimize the initial slot.

Our search method achieves the best result over a different number of iterations. The result is shown in Tab.3. The original simulated annealing algorithm does not bring benefits to the model. Instead, the model performance deteriorates significantly. The back-propagation algorithm approaches the performance of our method when the number of iterations is sufficient, which is too computationally

Table 2: Full model performance on CLEVR and ObjectsRoom. Models marked with red font are newly added during discussion.

| model | CLEVR | | ObjectsRoom |
|---|---|---|---|
| | ↑ARI-FG | ↓MSE | ↑ARI-FG |
| IODINE (Greff et al., 2019) | 93.81±0.76 | 44±9 | - |
| DTI (Monnier et al., 2021) | 89.54±1.44 | 77±12 | - |
| eMORL (Monnier et al., 2021) | 93.25±3.24 | 33±8 | - |
| MONet (Burgess et al., 2019) | 54.47±11.41 | 58±12 | 54±0 |
| GEN-v2 (Engelcke et al., 2021) | 57.90±20.38 | 158±2 | 84±1 |
| SA (Locatello et al., 2020) | 95.89±2.37 | 23±3 | 79±2 |
| I-SA (Chang et al., 2022) | - | 11±0 | 85±1 |
| BO-QSA (Jia et al., 2023) | 96.90±0.92 | 12±1 | 87±3 |
| AST-Seg-B3-BT (Jia et al., 2023) | 76.05±36.13 | 51±63 | 74.96±10.02 |
| SA+SLP (Jia et al., 2023) | - | - | 87±5 |
| BO-QSA+SLP (Jia et al., 2023) | - | - | **93±5** |
| RHGNet (ours) | 98.31±0.12 | 9±1 | 87±1 |
| +Infer-RHG | **98.55±0.09** | **8±1** | 88±0 |

Table 3: Comparison between different slot searching methods on CLEVRTex and CLEVR. We compare our searching method (**Infer-RHG**) with the simulated annealing algorithm (**sim-ann**) and back-propagation algorithm (**BP**) with different numbers of iterations.

| Iterations | Searching Method | CLEVRTex | | | | CLEVR | | | |
|---|---|---|---|---|---|---|---|---|---|
| | | ↑ARI-FG | ↑OIoU-S | ↑OIoU-M | ↑OIoU-L | ↑ARI-FG | ↑OIoU-S | ↑OIoU-M | ↑OIoU-L |
| 1(bottom-up) | - | 89.53 | 56.24 | 87.47 | 93.82 | 98.31 | 85.00 | 97.17 | 98.73 |
| 10 | sim-ann | 86.57 | 58.31 | 86.26 | 91.88 | 95.61 | 80.54 | 93.36 | 96.62 |
| | BP | 89.76 | 58.61 | 88.40 | 93.23 | 98.34 | 86.81 | 97.59 | 98.66 |
| | Infer-RHG | **89.90** | **58.91** | **88.76** | **93.88** | **98.55** | **87.47** | **98.02** | **98.89** |
| 100 | sim-ann | 87.27 | 59.05 | 87.14 | 92.75 | 96.45 | 82.47 | 94.42 | 97.42 |
| | BP | 89.92 | 58.48 | **89.01** | 94.43 | 98.45 | 86.53 | 97.73 | 98.80 |
| | Infer-RHG | **89.96** | **59.13** | 88.44 | **94.54** | **98.59** | **88.27** | **98.21** | **98.91** |

expensive. In contrast, our method achieves an acceptable result with ten iterations. Ninety more iteration steps bring only a slight performance improvement.

## C  GENERALIZATION TO REAL-WORLD SCENARIOS

Our method can easily be adapted to existing methods, thus generalizing to real-world datasets. Adding more informative signals, such as depth, optical flow, or pre-trained features is a common idea. Following the thought of DINOSAUR (Seitzer et al., 2022), we change the input and the reconstruction objective of the model from images to the output features of a DINO-pretrained Vision Transformer (Caron et al., 2021).

We give more detailed comparison here. The experiment results of MOVi-C and COCO are shown in Tab.4 and 5 respectively.

In Fig.2, we conduct visualization experiments on MOVi-C similar to those in the paper to prove our method also works on real-world scenes. We first repeat running our model with the image in Fig.2(b) as input 50 times, and record the relationship between Top-down Conflict and ARI-FG. The negative correlation between conflict and model performance also exists. Fig.2(b) illustrates how Infer-RHG works. Errorly segmented objects cause high conflict in their areas. Infer-RHG provides a

Table 4: Model performance comparison on MOVi-C, marked with the used pretrained backbone.

| model | pre-trained model | MOVi-C | |
| | | ↑ARI-FG | ↑mBO |
|---|---|---|---|
| Slot-Attention | - | 43.8±0.3 | 26.2±1.0 |
| SLATE | - | 43.6±1.3 | 26.5±1.1 |
| DINOSAUR | DINO ViT-S/8 | 67.2±0.3 | 38.6±0.1 |
| RHGNet | DINO ViT-S/8 | 70.50±0.3 | 39.64±0.2 |
| +Infer-RHG | | **73.00±0.3** | **40.68±0.2** |

Table 5: Model performance comparison on COCO, marked with the used pretrained backbone.

| model | pre-trained model | COCO | |
| | | ↑ARI-FG | ↑mBO |
|---|---|---|---|
| DINOSAUR | DINO ViT-B/16 | 40.5±0.0 | 27.7±0.2 |
| | DINO ResNet-50 | 36.0±0.5 | 22.9±0.4 |
| RHGNet | DINO ViT-B/16 | 41.02±0.5 | 27.90±0.4 |
| +Infer-RHG | | **41.14±0.4** | **28.26±0.4** |

larger probability of treating them as a whole object by resolving inconsistent boundary segmentation in object masks and top-down attention. In addition, Reverse Hierarchy Guidance takes effect on the low-level features. Based on DINO features according to Fig.2(c), the CNN encoder further highlights the foreground objects.

## D  FURTHER VISUALIZATION RESULT AND STABLE BACKGROUND SEGMENTATION

We provide further visualization results of our model to illustrate the reconstructed part of each slot, showing the object-centric property of our model. Fig.3, 4, 5, 6, 7 are the visualization results of CLEVR, ObjectsRoom, CLEVRTex, CLEVRTex-CAMO and CLEVRTex-OOD, respectively. Our model can extract the objects in the scene successfully in all the datasets. We also find that in ObjectsRoom, CLEVRTex, CLEVRTex-CAMO, and CLEVRTex-OOD, our model provides a stable background segmentation, where a separate slot reconstructs the images' background. However, this property is not observed in the CLEVR dataset. The background pixels are often reconstructed together with nearby objects. We consider it because the background of CLEVR barely changes. Thus the model does not need to learn semantic features for distinguishing the background pixels. In contrast, in other datasets, the background varies among images.

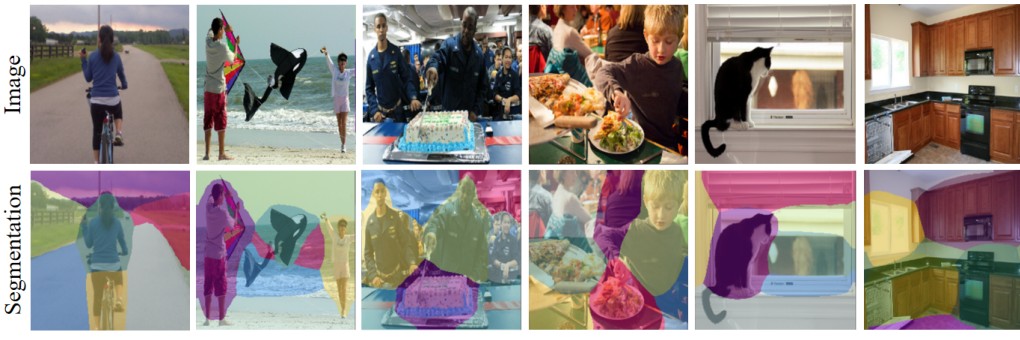

Figure 1: **COCO** visualization results.

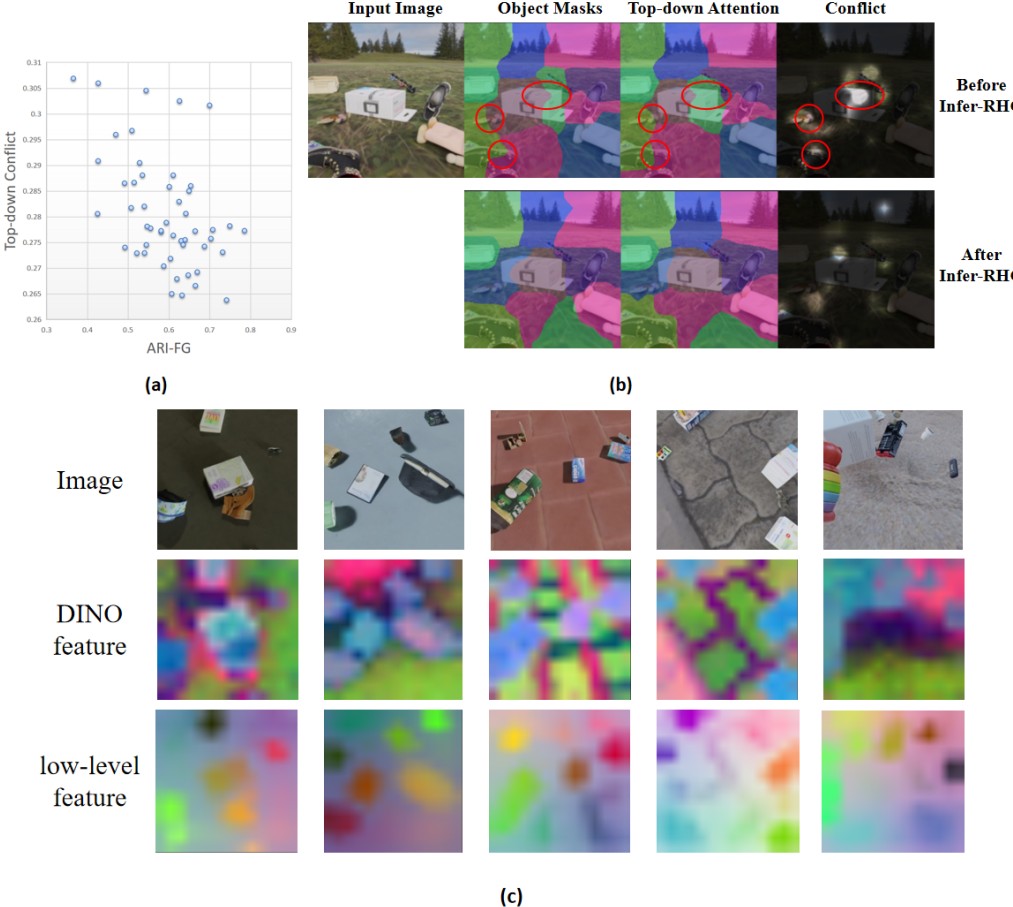

Figure 2: **MOVi-C** visualization. **(a)** Relation between top-down conflict and model performance, taking the image in (b) as input. **(b)** Influence of Infer-RHG on segmentation results. Conflicts often appear when an object is split into several parts due to inconsistent boundaries. **(c)** low-level feature visualization. Our approach highlights foreground objects based on DINO feature.

If it is necessary to segment the background pixels in CLEVR, a solution is to change the KL divergence $\mathcal{KL}(\cdot||\cdot)$ used to compute the top-down conflict to cross-entropy function $\mathcal{CE}(\cdot||\cdot)$. That is,

$$\mathcal{C}(\mathbf{F}, \mathbf{S}) := \mathcal{CE}(\mathbf{M}||\mathbf{A}) = -\sum_K \mathbf{M}\log(\mathbf{A}). \tag{1}$$

Minimizing the cross entropy function will bring two distributions close to each other and force them to be one-hot, thus forcing the background pixels to be assigned to a single slot. However, while this method enables the model to segment the background, it also reduces the model's ability to discover objects. We provide the visualization results in Fig.8, where quite a few small objects are ignored in the reconstructed images.

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

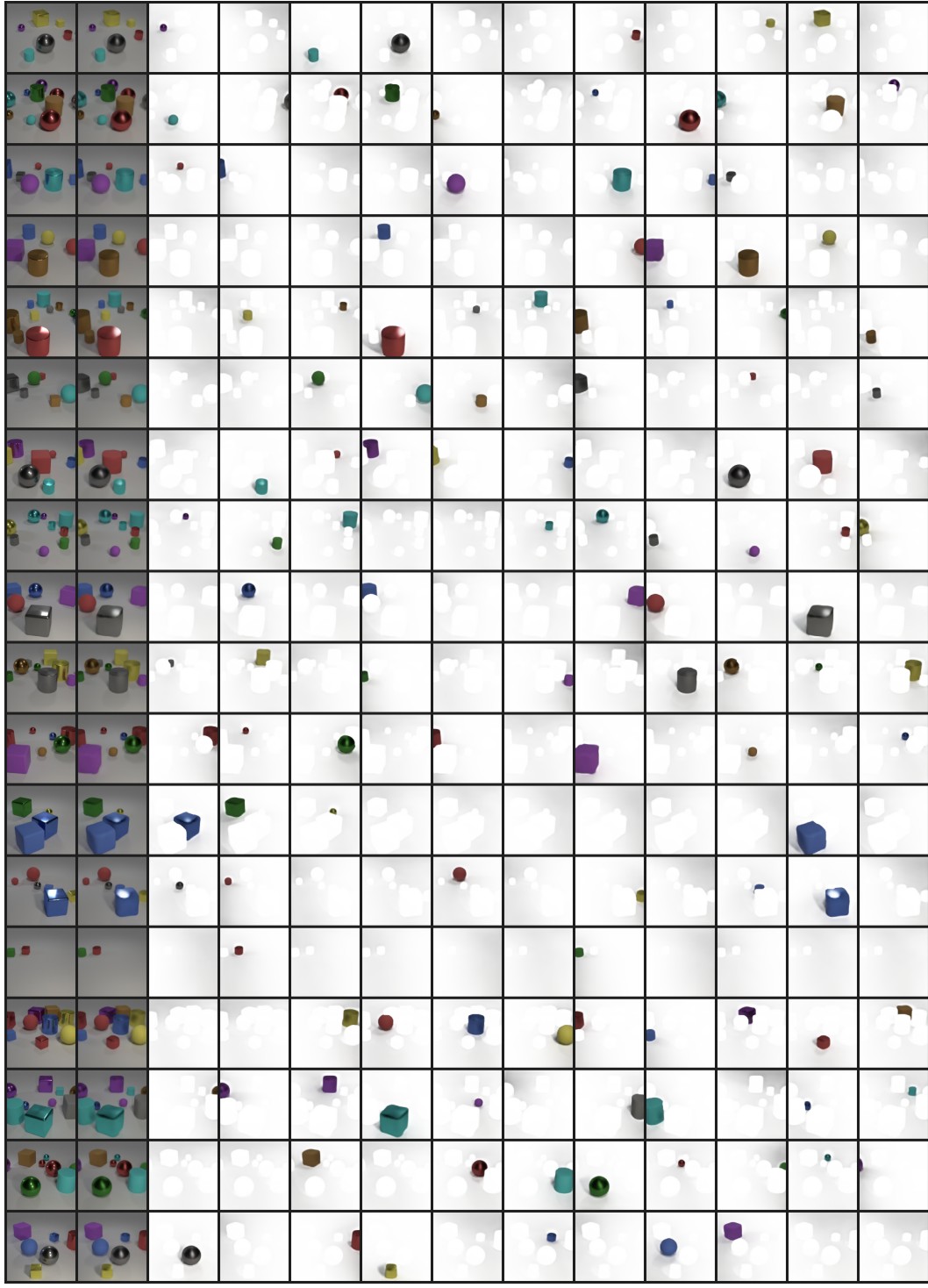

Figure 3: **CLEVR** visualization results. The first column is the input image, the second column is the reconstruction result, and the rest are each slot's reconstructed parts.

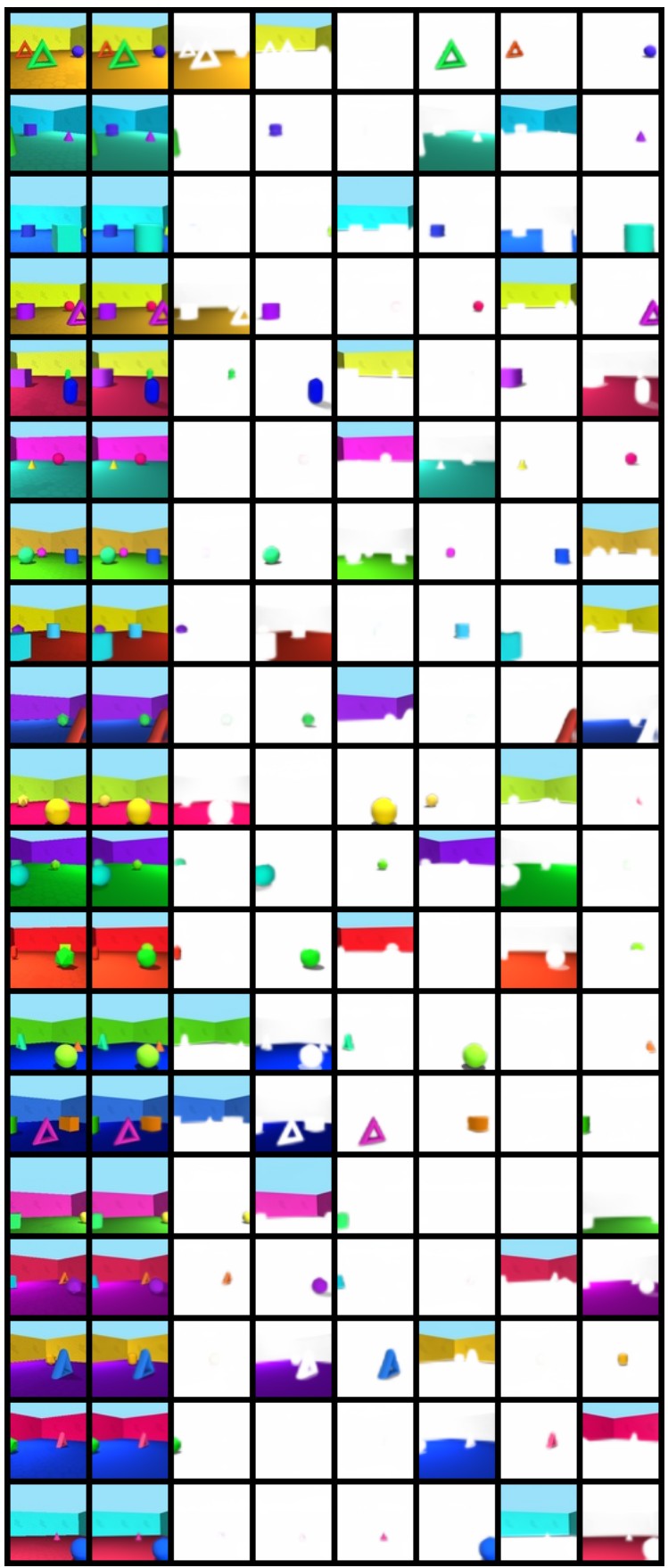

Figure 4: **ObjectsRoom** visualization results. The first column is the input image, the second column is the reconstruction result, and the rest are each slot's reconstructed parts.

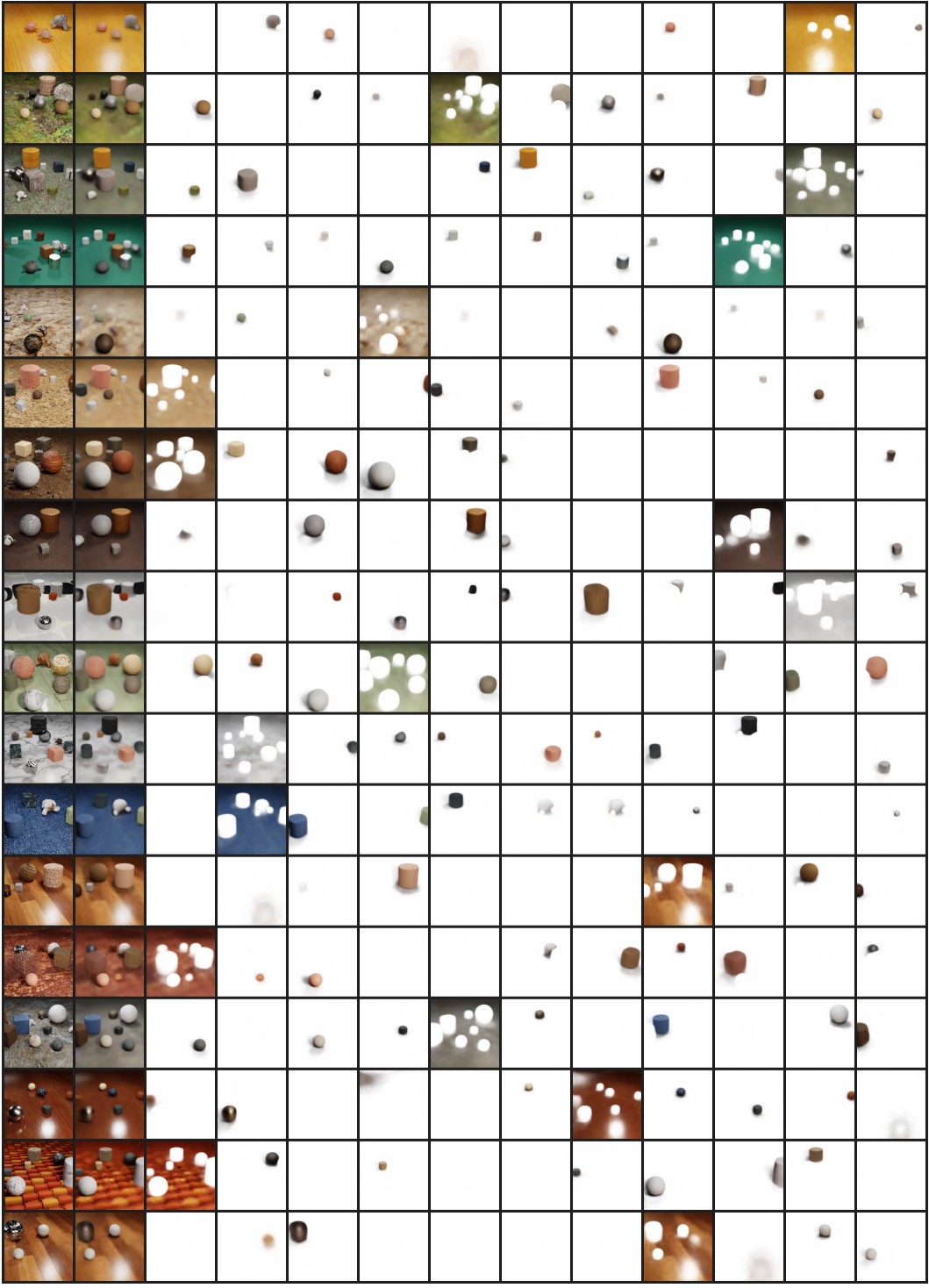

Figure 5: **CLEVRTex** visualization results. The first column is the input image, the second column is the reconstruction result, and the rest are each slot's reconstructed parts.

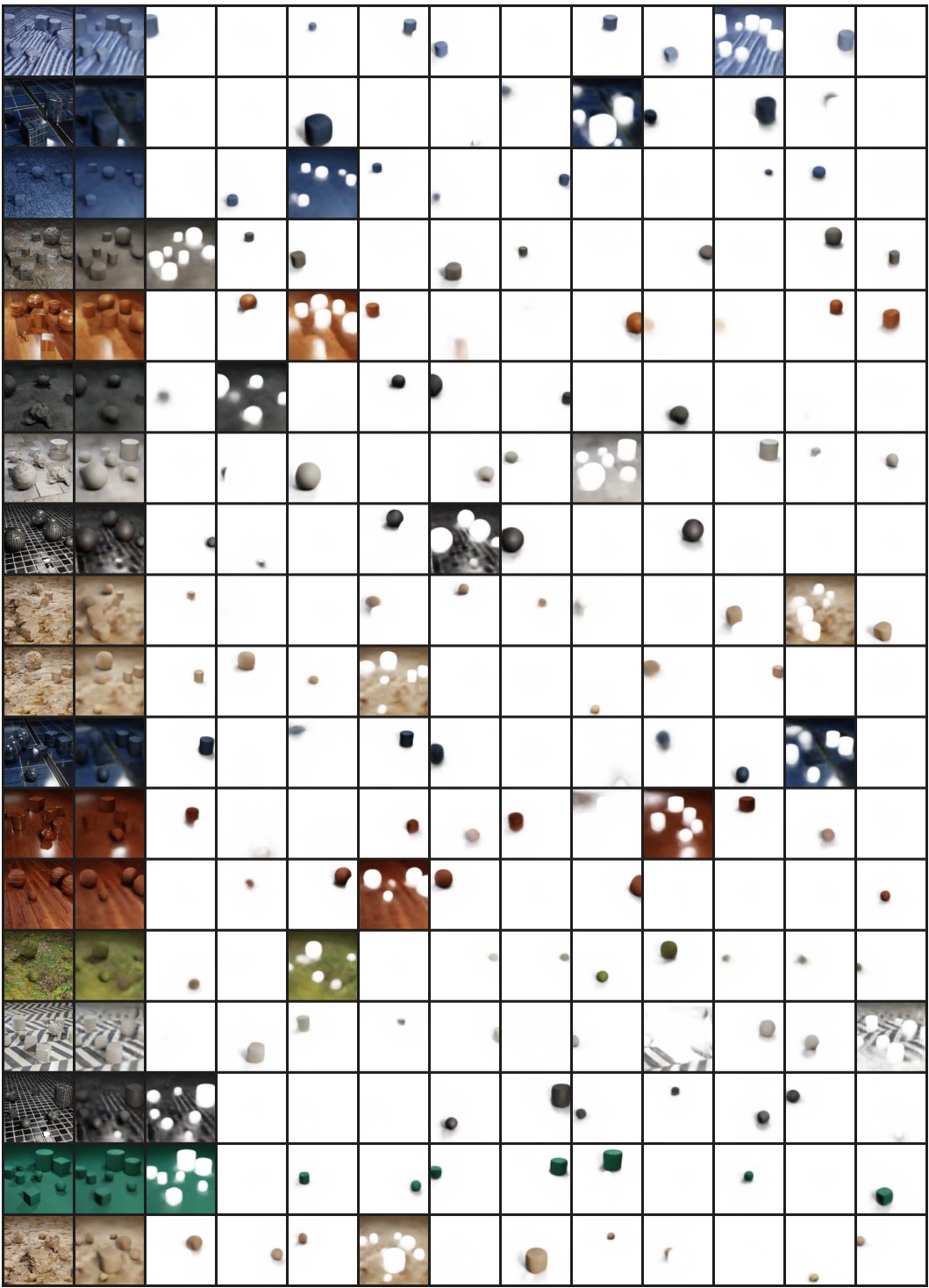

Figure 6: **CLEVRTex-CAMO** visualization results. The first column is the input image, the second column is the reconstruction result, and the rest is each slot's reconstructed parts.

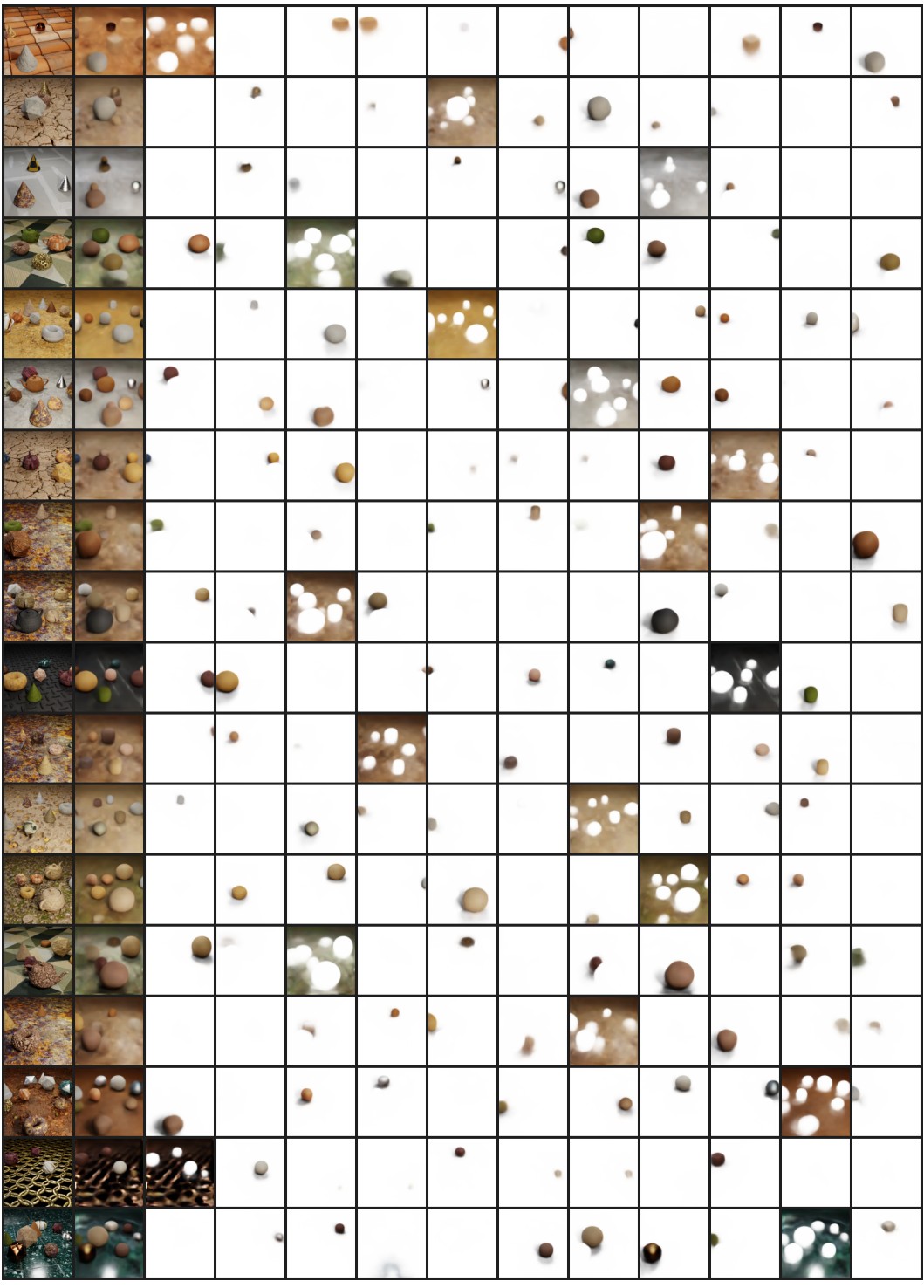

Figure 7: **CLEVRTex-OOD** visualization results. The first column is the input image, the second column is the reconstruction result, and the rest are each slot's reconstructed parts.

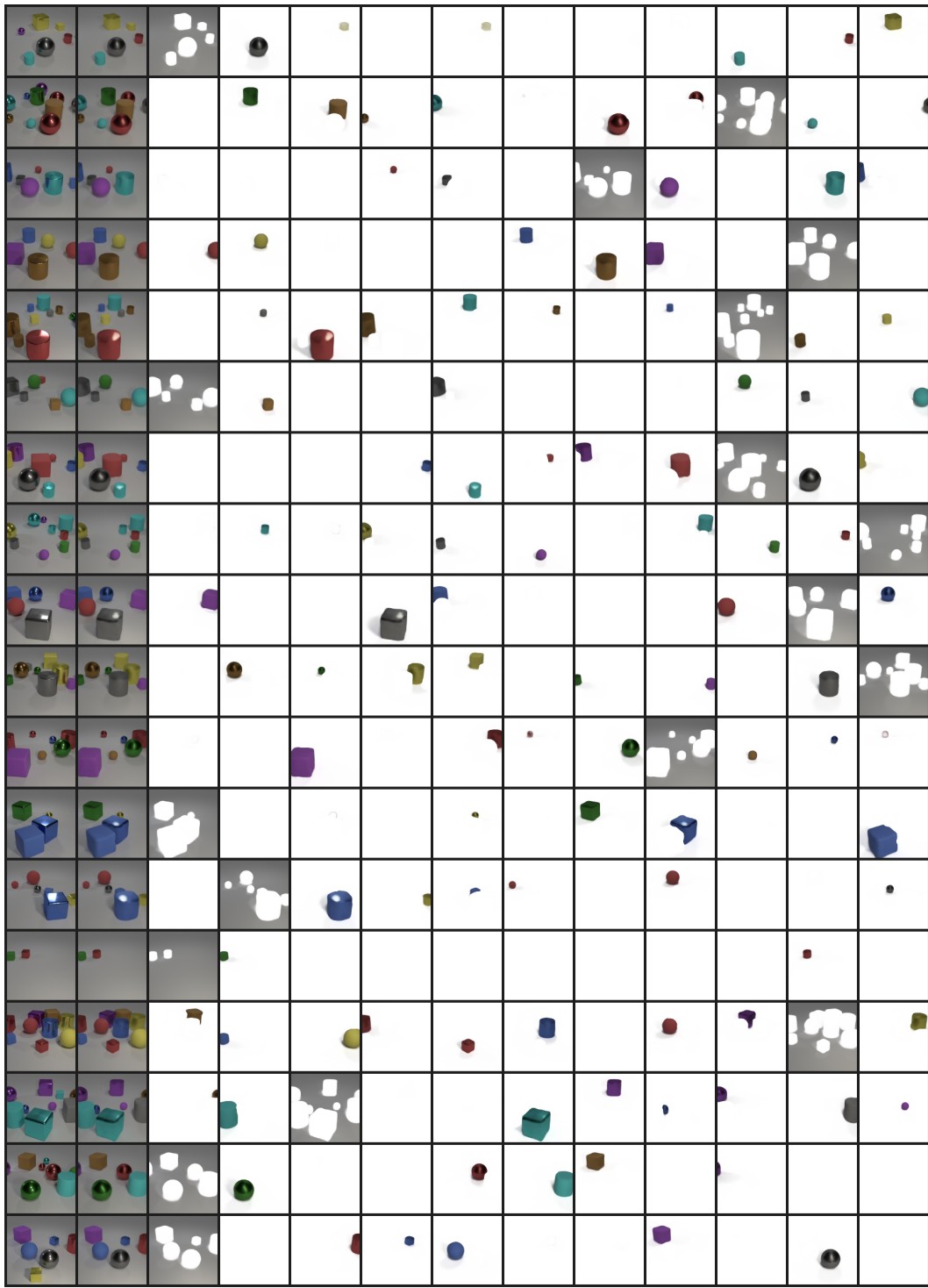

Figure 8: CLEVR model with a CrossEntropy top-down conflict. The first column is the input image, the second column is the reconstruction result, and the rest are each slot's reconstructed parts. The model successfully segments the background, but more objects are ignored compared with Fig.3, the result of the model with a KL divergence top-down conflict