# OpenReview forum: "Learning Object-Centric Representation via Reverse Hierarchy Guidance"
_ICLR.cc/2024/Conference — Submitted to ICLR 2024_

### Official Review · Reviewer_SoMm · 2023-10-30

**Soundness:** 3 good
**Presentation:** 3 good
**Contribution:** 2 fair
**Rating:** 5
**Confidence:** 4

**Summary:**

This paper proposes a method to improve on existing methods to parse scenes into object representations to better detect small/occluded objects. The method combines a bottom-up mechanism which tries to parse the scene into K object slots and a top-down mechanism which detects conflicts between the slot-based reconstruction and the original image and searches for a slot assignment which reduces conflicts. The approach gives improved results over benchmark algorithms on synthetic datasets.

**Strengths:**

The paper is mostly clearly written (there is some awkward grammar/wording) and makes excellent use of figures to illustrate results.

The proposed method is simple and effective.

There is a good amount of analysis and visualisation to explain the method.

**Weaknesses:**

The proposed method outperforms older SOTA models but does not clearly outperform some more recent work which incorporate different forms of “object” guidance into a bottom-up model:
https://openaccess.thecvf.com/content/WACV2023/papers/Sauvalle_Unsupervised_Multi-Object_Segmentation_Using_Attention_and_Soft-Argmax_WACV_2023_paper.pdf
https://arxiv.org/pdf/2305.19550.pdf
Given that these works involve some similar ideas to the current paper, it seems worthwhile to address them.

The section on "generalization to real-world scenarios" contains almost no information about what experiments were run or what the results were. If these results are important to evaluate the proposed method they should be explained in the paper.

**Questions:**

How does this method differ from other recent work with similar performance? What are the advantages of this proposed approach?

---

> ### Author Response · Authors · 2023-11-15
>
> Thank you very much for your review of our submission.
>
> ## About the model comparison in the paper
>
> We have included a comparison of the two models mentioned in Weakness in the updated version of the paper. It is important to note that all of our models on the synthetic dataset are trained from scratch. For a fair comparison, AST-Seg-B3-BT is added to the table. In addition, the strategy of SLP on CLEVRTex-OOD dataset is different from us: we train the model on the original CLEVRTex dataset and evaluate on CLEVRTex-OOD without finetuning, while SLP split a training set from the OOD dataset and trains their model on this training set. Due to different training conditions, we did not record their OOD performance. We list the updated comparison in the table below:
>
> |Model|CLEVRTex|ObjectsRoom|
> |:-:|:-:|:-:|
> |AST-Seg|71.79|75|
> |SA+SLP|71.00|87|
> |BO-QSA+SLP|87.00|93|
> ||||
> |RHGNet|89.53|87|
> |+Infer-RHG|89.90|88|
>
> ## Experiment on real-world datasets
>
> In our original paper, we mainly demonstrate the validity of our proposed method through experimental results on synthetic datasets. Experiments on real-world datasets are included in supplementary materials to prove that our method can be generalized to more complex scenarios. Thank you for your suggestions, we found that the generalization of the model in real-world scenarios is worth paying attention to, so we added the experimental data of the model in real-world scenarios in the updated paper. We list the ARI-FG comparison in the table below:
>
> |Model|MOVi-C|COCO|
> |:-:|:-:|:-:|
> |SA|43.8|16.4|
> |SLATE|43.6|24.1|
> |DINOSAUR|67.2|40.5|
> ||||
> |RHGNet|70.50|41.02|
> |+Infer-RHG|73.00|41.14|
>
> ## Difference between other Object-Centric methods
>
> Our main inspiration is to promote Object-Centric performance by incorporating some human visual system mechanisms into the network design. With an additional top-down pathway, we help the model learn more distinguishable features, as well as providing two alternative inference forms: a fast vision with a single bottom-up run, or a slow mode, running the network multiple times to get more accurate results. The experimental results show that our approach enables the model to learn more difficult samples, such as small or occluded objects in all the datasets, as well as texture-rich objects in CLEVRTex.

---

> > ### Comment · Reviewer_SoMm · 2023-11-22
> > **Response to authors**
> >
> > Thanks for the further information.

---

### Official Review · Reviewer_uwGV · 2023-10-30

**Soundness:** 3 good
**Presentation:** 2 fair
**Contribution:** 3 good
**Rating:** 6
**Confidence:** 4

**Summary:**

Inspired by Reverse Hierarchy Guidance, a theory related to human vision, the authors
propose an improvement for object-centric models based on clustering features into
slots. Central to the approach is measurement of conflicts between spatial feature
similarities and the clustering predicted by the model. The paper shows that established
object-centric models can be improved by minimizing conflicts during training and/or by
chosing the sampled segmentation with the least conflicts during inference. In
particular, the segmentation of small objects is improved which are more frequently
missed by existing methods.

**Strengths:**

- Existing methods are consistently improved on established, synthetic datasets.
- The authors provide an analysis that explains *why* the proposed method works by
  qualitatively and quantitatively inspecting segmentation performance for objects of
  different sizes. Beyond providing an improvemend model, the paper therefore also
  improves the understanding of object-centric modeling approaches.
- The Figures provided by the authors are helpful to understand the contribution. Beyond
  aggregated quantitative evaluation, the authors present figures that qualitatively
  showcase the improvements of the proposed method.

**Weaknesses:**

- In the main text, the proposed method is only evaluated on relatively simple,
  synthetic datasets. As shown in the supplement, the method can be combined with
  state-of-the-art object-centric models that scale to natural images, but the
  performance improvements do not seem to be significant.
- The mathematical description of the method in Section 3 is imprecise at several
  places. How exactly are the mappings $\mathcal{K}$ and $\mathcal{Q}$ defined? I.e.,
  what are the dimensionalities of the quantities involved in equation 3? In equation 4,
  the term in the summation does not depend on the summation index $K$.

**Questions:**

- The performance comparison in Table 1 uses FG-ARI to quantify segmentation
  performance. It has been pointed out several times in the literature that this metric
  is problematic (e.g., Engelcke et al 2020, Karazija et al. 2021, Monnier et al. 2021).
  How do the proposed methods perform in terms of the Object IoU metric?
- Does the Object IoU metric include evaluating IoU for the background segment?
- How were the object sizes chosen that where used to differentiate small, medium and
  large objects? Which fraction of the objects is small, medium and large, respectively?
  Beyond the agregated evaluation, it could be helpful to present a scatter plot of
  object size vs IoU for all objects in the evaluation set.
- How were the hyperparameters selected?
- Which performance is achieved when the features are clustered using a conventional
  approach such as normalized cuts instead of Slot Attention?

---

> ### Author Response · Authors · 2023-11-15
>
> Thank you very much for your review of our submission.
>
> ## Model Performance on Real-world Datasets
>
> In the supplemental material, the performance of DINOSAUR is directly taken from the report in [1]. However, DINOSAUR doesn't release public models. Therefore, we are not improving on their basis. Instead, we obtain a baseline model by simply changing the input and reconstruction objective of the network to DINO features, which achieve a bit lower performance than reported in [1]. This reduces the performance gains from our method.
>
> We have recently taken a closer look at the training tricks of previous work [1], for example, they reduce the dimensions of pre-trained features to lower dimensions at the input side and recover to the original dimensions at the output side. We find that these modifications can improve the clustering performance of the network with less computation. We apply these tricks to our model and retrain the model on real-world datasets. In the table below we provide a comparison of model performance before and after retraining.
>
> |Model|MOVi-C|COCO|
> |:-:|:-:|:-:|
> |DINOSAUR|67.20|40.50|
> ||||
> |original reproduction|65.68|37.90|
> |+Train-RHG|67.48|40.57|
> |+Infer-RHG|70.88|40.80|
> ||||
> |new reproduction|67.08|38.74|
> |+Train-RHG|70.50|41.02|
> |+Infer-RHG|73.00|41.14|
>
> After more thorough training of the network, we can come to two conclusions: (i) our performance benefits from better network settings, reaching +5.80% and +0.64% ARI scores compared with DINOSAUR; (ii) Compared with our reproduced baseline model, both Train-RHG and Infer-RHG bring remarkable performance gains.
>
> ## About Object-IOU metric
>
> In the supplementary material, we present the comparison of our method and other models in terms of Object IoU in CLEVRTex (Table 6 in supplementary material), where our method still outperforms other SOTA methods compared in the paper. The computation of Object IOU doesn't consider background pixels because we focus on the ability of the model to distinguish objects.
>
> ## About object size division
>
> We divide objects by their size distribution. Specifically, we count the number of pixels that all objects occupy in the ground truth, and we classify objects into small, medium and large on a ratio of about 1 to 5 to 1.
>
> A scatter plot of object size vs IoU is a better way to express how the model's ability to discover objects varies with the size of objects. According to your suggestion, we have redrawn a plot and put it in the latest update, you can check it in the updated paper.
>
> ## About hyperparameter choosing
>
> Most of the hyperparameter settings follow previous work [3]. The main hyperparameter discussed in this paper is the weight of top-down conflict loss. The weight is 0 at the beginning of the training and gradually increases to a maximum value. This setup follows the intuition that the model needs to first learn to distinguish objects, and then enhance low-level features through top-down guidance to gain a stronger ability to discover objects.
>
> The maximum weight of top-down conflict loss depends on the complexity of datasets. For simple scenarios, such as CLEVR, smaller weights are appropriate because increasing the weights will lead to a decrease in the accuracy of model reconstruction and segmentation. For more complex datasets, such as the CLEVRTex dataset, it is more necessary to find missing objects than to get more accurate segmentation, thus requiring a larger top-down conflict loss weight.
>
> ## Mathematical description and typo error
>
> We'll add a more detailed description of the symbols in section 3. The mapping $\mathcal{K}$ and $\mathcal{Q}$ doesn't change the dimensionality of features, still retaining $C=64$ as described in section 4.1. In addition, the summation index $K$ in Equation 4 appears due to a typo error, and it should be removed. In the updated paper, we have corrected these problems.
>
> ## Conventional clustering approach
>
> Considering the similarity of slot attention and k-means algorithm, we use k-means to cluster the low-level features output by the CNN encoder. We take the CNN encoder from two networks, namely RHGNet and BUNet in the paper, and cluster them respectively. After clustering the features of RHGNet, we obtain an ARI-FG score of 52.29%, while the clustering score of BUNet is 41.42%. The results verify that RHGNet learns better features.
>
> [1] Bridging the gap to real-world object-centric learning
>
> [2] Self-supervised Object-Centric Learning for Videos
>
> [3] Object-Centric Learning with Slot Attention

---

> > ### Comment · Reviewer_uwGV · 2023-11-17
> >
> > Thank you for your response and the revision of the paper. Most of my points have been addressed.
> >
> > However I am confused by the newly added Figure 6, which shows the object IoU for different object sizes. To me these results do not show a clear performance improvement for small objects. In particular when comparing to BO-QSA, the performance of RHGNet seems to be only better for larger objects (>=300). Could you elaborate on this discrepancy to the qualitative results you discuss in the paper?
> >
> > For clarification, how to you measure object size? Is it the number of pixels?

---

> > > ### Author Response · Authors · 2023-11-18
> > >
> > > We use the number of pixels occupied by an object in the ground truth as the object size.
> > >
> > > As we claimed in the paper, RHGNet overcomes more challenging samples that a bottom-up model can not solve. Due to the difficulty of CLEVRTex, the existing models have frequent errors on objects of all sizes, and RHGNet has performance improvements on all sizes. On the relatively simple datasets, RHGNet's advantages on small objects are better reflected. We have replaced Figure 6 with the comparison between RHGNet and BOQSA on CLEVR. In terms of OIoU, RHGNet has achieved absolute performance advantages on objects smaller than 200 pixels.

---

> > > > ### Comment · Reviewer_uwGV · 2023-11-22
> > > >
> > > > Thank you for clarifying. The updates during the discussion phase clearly improved the paper in my view, therefore I am happy to increase my rating.

---

### Official Review · Reviewer_TuCC · 2023-10-31

**Soundness:** 3 good
**Presentation:** 2 fair
**Contribution:** 2 fair
**Rating:** 6
**Confidence:** 3

**Summary:**

The paper's major contribution is introducing idea of top-down modulation in Reverse Hierarchy Theory to the standard slot-attention model aimed for object-centric learning. The major architecture novelty is  a top-down path from the slots of slot attention model to attend the image features and directly generate an attention map by a CNN decoder. This attention map is in turn used to compute a conflict map against the original object mask produced directly by the slot representation through a spatial broadcasting. The conflict loss appears to be the major driving force for the model to improve its ability of discovering small objects, which other models often fail to detect. The model demonstrates superior performance to other models on several benchmark datasets with pure colors, but also generalizes to CLEVRTex which introduces some texture within each object with a dominant distinct color.

**Strengths:**

(1) All the benchmark result show that the proposed model outperform previous models.

(2) The idea of conflict loss between the two object mask maps appear novel.

(3) Analysis of the representation of objects in CLEVRTex dataset shows that the model learned a representation that has smaller variance in intra-object feature (locked to image grids) than inter-object feature, which may help better group pixels within an object into a same slot. Such analysis is helpful for partially understanding why a model learns better (but I still remain somewhat puzzled, please see my comment in Questions)

**Weaknesses:**

(1) At inference, the model needs to be run for multiple times to get a performance gain. This increases computational time.

(2) I think it is worth acknowledging some limitation observed in the visualization. One of the examples in ObjectsRoom of Figure 4 actually shows the proposed model is the only one that hallucinates the yellow color of the floor seen through the hollow part of the triangle shape. From the supplementary material, it seems often "fill in" the hollow part o the triangle in the segmentation mask, which should actually be background. From visual inspection, it seems that for the more challenging datasets such as CLEVRTex, missing small and partial objects is still frequent. And the model appears to often ignore the detailed texture in reconstruction but only captures the average color. Of course all models have limitations, I think it is worth pointing these out in discussion, and it will be interesting to postulate why texture often gets ignored.

(3) It is more of a question but also a weakness (only for the sake of gaining insight): I still lack full understanding of where the teaching signal or inductive bias comes from to get the improvement. See my comment (1) in Questions.

**Questions:**

(1) Although conceptually it makes sense that top-down modulation should help in object perception, because obviously the brain uses it, the computational mechanism of why it helps in the experiments presented in the paper still remains puzzling to me. I am happy to see the analysis in Figure 7 of the intra-object vs. inter-object feature variance and Figure 6 illustrates the failure of bottom-up only network. But my question is: what even caused these improvement. My guess is that "in-principle", when the top-down attention from the slot includes some aggregated information over pixels of the objects, it is possible for the low-level features to be biased towards such aggregated information. But "in-principle" does not mean this is guaranteed. Moreover, what drives the network being able to learn to detect small objects better? It sounds that the paper indicates that the conflict loss term is the guidance, but the conflict term is simply the conflict between two masks that are both internally generated and to be learned, with no additional inductive bias or teaching signals by introducing the top-down pathway.

To illustrate my point in more detail: if the slot representation has already lost one small object in the object masks M as in Figure 2, why would it necessarily produce the mask of the missing object through the attention to low-level feature and eventually show in the attention map A? Why would not minimizing the conflict loss C drive A to be more close to M (which will remove the small object) instead of vice versa? There seems to be no teaching signal directly from the image to constrain the attention map A, so I would guess that A is initially random in early stage of learning and is taught by M. Is that the case? If indeed A starts being better than M (it includes the missing objects while M does not), then why not just include A alone in the model? Why bother introducing M?

One guess is that the major contributor for the better ability of detecting small objects is just the extra depth in the attention map pathway introduced by the top-down attention. If so, would introducing more iterations in the original slot attention model achieve similar improvement? I wonder if the authors can elucidate the actual mechanism.

(2) I would like to get a confirmation that no component of the network is pretrained on other tasks. If some of them are pre-trained, please explain.

(3) At inference time, the model is run N times and the one with the smallest conflict C is chosen. I wonder whether the performance improvement mainly comes from this selection process. What happens to the performance if you set N to 1? I wonder if the authors think that this variation has some similarity to the brain? Perhaps the brain does not always detect all objects with one (or two) glances. The ones being missed by the brain depends on what a person's initial top down attention is on.

---

> ### Author Response · Authors · 2023-11-15
>
> We appreciate your review of our submission.
>
> ## Pretraining confirmation
>
> All the models on synthetic datasets are trained from scratch without using pretraining weight. Models on real-world datasets use the output feature of DINO to provide reconstruction objectives, and the pre-trained models have been indicated.
>
> ## Weakness 1 & Question 3: Further explanation about Infer-RHG
>
> The network can either quickly extract objects in the scene through a pure bottom-up inference, or it can run the network multiple times and search for a more precise perception through a top-down pathway. When N is set to 1, the network only runs the bottom-up process, which does not require more computational complexity, but still outperforms other SOTA models. When the top-down path is started during inference (Infer-RHG in the paper), the network requires more computation, and the performance will further increase. In the paper, every time we compare performance with other models, we list two performances that correspond to situations where N is set to 1 and 10 respectively.
>
> The performance improvement of Infer-RHG comes from the selection of initial slot values. Random initial values in slot-attention will produce different attention distributions, eventually leading to performance differences, as shown in Figure 3 in the paper. This is similar to human visual perception, where sometimes we don't pay attention to objects with low saliency. Our model obtains multiple attention distributions through different initial slots so that these objects have a greater chance of being found. Using Top-down conflict as a signal, we pick out the perception that is most likely to find these objects, thus discovering small objects.
>
> ## Weakness 2: RHGNet focuses more on object discovery ability
>
> The top-down pathway demands the network to retain more general features because slots need to match all the features in the area they occupy, which causes slots to tend to reconstruct an "average" object that represents the overall features of the area while discarding more detailed features such as textures. This sometimes causes the model to converge to less precise reconstructions, but this doesn't always appear. We examined three models on ObjectsRoom trained with different random seeds. The model used for display had the problem described in Weakness, while the other two performed correctly, and all three models had an ARI score of 87% to 88%. We have updated the paper and replaced these figures with a correctly reconstructed version.
>
> In addition, note that our displayed images are not carefully selected, but are directly chosen by taking out the first several images from test sets, thus the displayed result represents the average performance of our model. Among the 18 images shown in the supplementary material, a total of 107 objects appear, and our model misses 4 of them, which has made improvement compared with the current models shown in Figure 4 in the paper.
>
> ## Weakness 3 & question 1: Working mechanism of our method
>
> Conflict loss just drives $A$ to get closer to $M$. We add Information Integrity Loss $L_\mathrm{info}$ to avoid the situation that the network learns to decrease conflict loss by removing an object from low-level features. We use a shallow decoder to reconstruct the image from output features of the CNN encoder, and $L_\mathrm{info}$ is the reconstruction loss. This loss guarantees the existence of all objects in low-level features.
>
> $A$ is initially random and is meaningless at this time, thus we set the weight of conflict loss to 0 at the beginning. Using only reconstruction loss, slot attention can discover objects in a simple scene in the form of object mask $M$. This preliminary perception may miss objects, but it is at least "broadly correct". We think of $M$ as a pseudo label and use it to supervise the CNN encoder: it learns a segmentation task through conflict loss. Despite given annotations with some errors, the encoder can still effectively learn object concepts, which in turn makes slot attention clustering easier, thus obtaining better object discovery results.
>
> The authors of [1] experimented with the number of slot attention iterations during training, finding that more iterations may decrease performance. We list experimental data on the number of slot attention iterations from their paper in the table below:
>
> |Iterations|ARI-FG on CLEVR6|
> |:-:|:-:|
> |1|85.2|
> |2|98.0|
> |3|98.8|
> |4|98.7|
> |5|92.0|
>
> This suggests that the depth of the network is not necessarily beneficial for learning Object-Centric representations. Our top-down path does not introduce additional depth to the network, but rather introduces a shortcut that directly connects the perception of the upper level (slot) with the lower level (CNN features), thus benefiting the learning of low-level features.
>
> [1] Object-Centric Learning with Slot Attention

---

### Official Review · Reviewer_UGEP · 2023-11-03

**Soundness:** 3 good
**Presentation:** 3 good
**Contribution:** 2 fair
**Rating:** 5
**Confidence:** 3

**Summary:**

This paper proposes to add a top-down attention mechanism in current standard Object Oriented Learning models to overcome a blindness issue that occurs for occluded and small objects in crowded scenarios.

**Strengths:**

The method introduced for using the top down guidance adds a loss function during training that is based on minimising a consistency measurement that evaluates the match of low features extracted with the object slots using KL divergence between these two factors.
At inference time, the number of iterations for the refinement with the top down consistency approach has an impact on the final performance, and it might be a nice way to having a compromise between inference computational time and final performance.

A thorough experimental set-up with ablation studies is reported, demonstrating the higher accuracy of the model in comparision to state-of-the art and the contribution to each of the components.

**Weaknesses:**

The connection to human visual system is rather weak, and even though it can serve as inspiration, there is no evidence that this could be a computational model for human computations of top down signals. There are other works that point to recurrent connections and other mechanisms for human brain modeling.

**Questions:**

See weaknesses points

---

> ### Author Response · Authors · 2023-11-15
> **Connection between RHGNet and human visual system**
>
> Thank you very much for your review of our submission.
>
> We do not directly simulate the internal structure of human visual system, but rather achieve its function. In our main reference theory, Reverse Hierarchy Theory, the authors describe two main functions of human visual system, namely "Vision at a glance" and "Vision with scrutiny". The bottom-up and top-down pathways are responsible for the two kinds of vision. In our network, the top-down path will generate a conflict signal allowing the network to evaluate its output. The network selects a perception that minimizes the signal from multiple perceptions, thus obtaining as correct a perception as possible. At the same time, this process is optional, the model can not only obtain information quickly through pure bottom-up inference, but can also apply the top-down pathway for more careful observation and correcting perceptual errors. So, functionally, our networks are closer to the human visual system.

---

### Meta-Review · Area_Chair_JZHB · 2023-12-05

**Metareview:**

This paper proposes to align encoder-based and decoder-based masks in Slot Attention to achieve improved unsupervised scene decomposition performance. It relies on attention masks obtained in the encoder and composition masks obtained from a spatial broadcast decoder and introduces a loss to align the two. The paper further draws connections to Reverse Hierarchy Theory (RHT) and describes this alignment process as a top-down process. Finally, it introduces a model ensembling heuristic for running the model forward pass multiple times at inference time to select (without supervision) the best performing instantiation. Together, these techniques provide improvements on standard evaluation metrics for across several typical benchmark tasks in unsupervised scene decomposition.

Overall, the reviewers agree that this is a novel and interesting contribution to a growing literature on unsupervised scene decomposition / object-centric learning.

Several concerns are highlighted: 1) the motivation to particularly improve on small objects is only partially verified (although, I believe, Figure 6 is a step in the right direction), 2) the connection to Reverse Hierarchy Theory (RHT) is very loose, and 3) the mathematical notation is imprecise.

After reading the paper myself (as the reviewers couldn’t agree on a accept or reject recommendation), I noticed several additional issues with the work that I believe require a larger revision before the paper can be accepted:

1) On the synthetic datasets like CLEVRTex, CLEVR etc., the authors use a ResNet-18 backbone whereas prior related work uses simpler backbones. A recent paper, Invariant Slot Attention [1], showed that the choice of backbone can make a substantial difference in performance, particularly on CLEVRTex – significantly outperforming all the reported numbers in RHGNet. Even a baseline Slot Attention model, when trained w/ a ResNet backbone, does significantly better. This is a major concern that requires careful re-framing of the work. For the COCO experiments, where the same backbone is used as in prior work, the performance difference is no longer significant (excl. the Infer-RHG method, see comment below). There may be a small benefit on the synthetic MOVi-C dataset, but it is difficult to judge in the presence of the other results.

2) The “Infer-RHG” model variant effectively adds test-time optimization to use additional compute to improve model performance. Given recent work on Slot-TTA [2] it is not very surprising that test-time optimization can give you performance improvements (at the expense of inference speed). The particular scheme that is proposed in this paper is interesting, but it would require more careful comparison against other test-time optimization methods to understand its trade-offs (e.g. whether it is more compute-efficient).

Overall, I am convinced that these two concerns (in addition to the overall feedback by the reviewers) bar the paper from qualifying for acceptance at ICLR. The ideas presented in the paper are very interesting and I hope the reviewers can carefully address these two points when preparing a future version of this work.

[1] Biza et al., Invariant Slot Attention: Object Discovery with Slot-Centric Reference Frames (ICML 2023)
[2] Prabhudesai et al., Test-time Adaptation with Slot-Centric Models (ICML 2023)

**Justification For Why Not Higher Score:**

Inadequate comparison against prior / related work; unclear if benefits are attributable to model improvements or simply based on using a stronger backbone.

**Justification For Why Not Lower Score:**

N/A

---

### Decision · Program_Chairs · 2024-01-16

Reject